# Role of PQBP1 in Pathogen Recognition—Impact on Innate Immunity

**DOI:** 10.3390/v16081340

**Published:** 2024-08-21

**Authors:** Lukas Wiench, Domenico Rizzo, Zora Sinay, Zsófia Nacsa, Nina V. Fuchs, Renate König

**Affiliations:** Host-Pathogen Interactions, Paul-Ehrlich-Institut, Paul-Ehrlich-Str. 51–59, 63225 Langen, Germany

**Keywords:** PQBP1, innate immunity, neurodegeneration, Renpenning syndrome spectrum

## Abstract

The intrinsically disordered polyglutamine-binding protein 1 (PQBP1) has been linked to various cellular processes including transcription, alternative splicing, translation and innate immunity. Mutations in PQBP1 are causative for neurodevelopmental conditions collectively termed as the Renpenning syndrome spectrum. Intriguingly, cells of Renpenning syndrome patients exhibit a reduced innate immune response against human immunodeficiency virus 1 (HIV-1). PQBP1 is responsible for the initiation of a two-step recognition process of HIV-1 reverse-transcribed DNA products, ensuring a type 1 interferon response. Recent investigations revealed that PQBP1 also binds to the p17 protein of avian reovirus (ARV) and is affected by the ORF52 of Kaposi’s sarcoma-associated herpesvirus (KSHV), possibly also playing a role in the innate immune response towards these RNA- and DNA-viruses. Moreover, PQBP1-mediated microglia activation in the context of tauopathies has been reported, highlighting the role of PQBP1 in sensing exogenous pathogenic species and innate immune response in the central nervous system. Its unstructured nature, the promiscuous binding of various proteins and its presence in various tissues indicate the versatile roles of PQBP1 in cellular regulation. Here, we systematically review the available data on the structure of PQBP1 and its cellular functions and interactome, as well as possible implications for innate immune responses and neurodegenerative disorders.

## 1. Introduction

PQBP1 was discovered as a binding protein in polyQ tract sequences [1]. It is well known as a pre-mRNA splicing factor [2], with an important role in transcriptional [3] and translational regulation [4], and is expressed in various tissue types and widely distributed in different brain regions [3,5,6]. Recently, PQBP1 has been identified as an indispensable cellular factor of a priming event for recognition of HIV-1 infection by the cGAS–STING pathway, accentuating PQBP1’s role in inflammation and in innate immune response regulation [7].

Mutations in the *PQBP1* gene have been identified in patients suffering from different disorders, including Sutherland–Haan syndrome, Golabi–Ito–Hall syndrome, Hamel cerebro-palato-cardiac syndrome, Porteous syndrome and MRX55, which are collectively called the Renpenning syndrome spectrum [8,9,10,11]. The *PQBP1*-linked syndromes share common clinical features including microcephaly, intellectual disability (ID), reduced growth, lean body and muscular atrophy. First symptoms begin to appear in the early years of childhood [8]. While the role of PQBP1 in the pathogenesis of neurological disorders has not been fully elucidated, many studies have identified functions of PQBP1 that might explain the symptoms appearing in Renpenning syndrome spectrum patients. PQBP1 was also identified interacting with the causative proteins of polyQ diseases such as Huntington’s disease, Kennedy’s disease and spinocerebellar ataxia [6,12,13], as well as with tau proteins [14], thereby suggesting a role for PQBP1 in neurodegenerative disorders. In addition, PQBP1 has been associated with both cancer and cardiovascular diseases [15,16,17,18].

In this review, the effects of disease-associated *PQBP1* variants and mutations will be examined, and we will provide an overview of the current knowledge on the functions of PQBP1 in innate immunity and correlation between inflammation and neurodegeneration.

## 2. PQBP1 Structure and Functions

PQBP1 is an intrinsically disordered protein (IDP) 265 amino acids in length [19]. IDPs are characterized by the presence of low complexity and repetitive domains with a low content of hydrophobic amino acids, which normally are found in the core of folded globular proteins. In addition, they contain a high proportion of polar and charged amino acids [20]. Unstructured regions are quite common in functional proteins [21], in particular in those involved in nucleic acid binding, crucial for transcriptional regulation, translation and cellular signal transduction [22,23]. PQBP1 harbors five confirmed domains: the N-terminal domain (NTD), a WW domain (WWD), a polar-amino-acid-rich domain (PRD), a nuclear localization signal (NLS) and a C-terminal domain (CTD) [3,6]. For each of these domains, various interaction partners have been described. Takahashi et al. suggested that these interactions might be strongly regulated by the unstructured nature of PQBP1 [24]. They also proposed that PQBP1’s structure allows PQBP1 binding to two partners simultaneously. The lack of structure and the high flexibility enable PQBP1 to promiscuously bind various proteins (see Figure 1), easily overcoming steric restrictions by adopting different conformations [24].

### 2.1. Mapped PQBP1–Protein Interactions

The NTD of PQBP1, comprising residues 1 to 46, forms the site of association with an incoming HIV-1 capsid (CA) upon infection of a cell. It recognizes an electropositive pore in the center of HIV-1 capsid hexamers/pentamers, which is formed by arginine-18 residues of six/five CA subunits, and is highly conserved among lentiviruses. Binding of PQBP1 to CA is the first interaction required for the initiation of an innate immune response during HIV-1 infection [7].

The WWD (residues 48–81) is the only part of the PQBP1 protein that has a strong secondary structure containing a triple-stranded anti-parallel β-sheet [25]. As members of the SH3 domain family, WWDs are well known for their ability to bind specifically to proline-rich sequences, which are extensively distributed in both cytoplasmic and nuclear proteins, enabling putative interactions with a variety of candidates [26,27]. In the case of PQBP1, pre-mRNA splicing factor WW domain-binding protein 11 (WBP11) was the first protein that was confirmed to bind to the WWD [28]. In 2020, Park et al. showed that in human cell lines, WBP11 regulates the splicing of 164 genes, which are not enriched for any biological pathway or cellular function [29]. In the same study, it was shown that WBP11 regulates centriole duplication during cell proliferation by affecting splicing of TUBGCP6 pre-mRNA [29]. Additionally, Iwasaki and Thomsen showed that both PQBP1 and WBP11 are expressed in the nascent mesoderm and neuroectoderm and are crucial for the proper development of mesoderm and neural plate during early development of *Xenopus laevis* embryos [30]. Llorian et al. hypothesized that PQBP1 was in part responsible for the translocation of WBP11 to the nucleus [31]. While the functions of WBP11 are still being elucidated, there is an indication that proper function of WBP11 influences many crucial cellular processes, and that its association with PQBP1 requires further investigation. Regarding to its role in splicing, Wang et al. reported that depletion of PQBP1 induced the expression of proapoptotic Bcl-xS in favor of anti-apoptotic Bcl-xL through an increase in the use of an alternative 5′ splice site. In addition, they reported alternative splicing of numerous mRNAs particularly involved in neurite outgrowth in primary murine neurons [32]. Renpenning syndrome disease-linked frameshift mutations in the PRD, and particularly a missense mutant with a point mutation in the WW domain (Y65C), lost association with splicing-related factors, e.g., the central U2 small nuclear ribonucleoprotein (snRNP) component SF3B1, suggesting that the complex of SF3B1 and PQBP1 influences splicing decisions on a subset of mRNAs [32]. The role of PQBP1 in pre-mRNA splicing was emphasized by identifying its interactions with additional spliceosomal proteins such as U5-15kD and U5-52kD, which will be described later [33].

WWD not only plays a role in alternative splicing but it also directly interacts with the CTD of RNA polymerase II, suggesting that PQBP1 plays a regulatory role in transcription [12]. The interactions of PQBP1 with RNA polymerase II and spliceosomal proteins strongly indicate the influence PQBP1 exerts on transcription and pre-mRNA splicing. RNA polymerase II is regulated in a phosphorylation-dependent manner, since its CTD is rich in potential phosphoacceptor amino acid residues [34]. Okazawa et al. hypothesized that phosphorylation in the CTD sequence of RNA Pol II could increase the binding affinity to PQBP1. Interestingly, the authors found that the level of phosphorylation of RNA polymerase II appeared to be reduced when mutant ataxin-1 was co-expressed with PQBP1, which also reduced overall transcription level, suggesting a PQBP1 involvement in RNA Pol II phospho-regulation as well [12].

In 2021, Shen et al. described PQBP1’s role in the regulation of translation elongation. They demonstrated that the WWD of PQBP1 associates with eukaryotic elongation factor 2 (eEF2), a member of the GTP-binding translation elongation factor family. Phosphorylation of eEF2 reduces the translational elongation, resulting in decreased protein synthesis. PQBP1 binds with a higher affinity to eEF2 when eEF2 is not phosphorylated at Thr56, thereby preventing the phosphorylation of Thr56. Thus, PQBP1 induces protein synthesis by preventing phosphorylation of eEF2 [4].

In addition to its important roles in splicing, transcription and translational regulation, the WWD of PQBP1 has also been found to interact with cyclic GMP–AMP (cGAMP) synthase (cGAS) [35] and dynamin-2 [36].

cGAS is one of the most important DNA sensors, responding to cytosolic DNA and activating type I interferon responses [37]. A disease-associated mutant (Y65C) revealed that PQBP1 requires an intact WW domain for binding cGAS. This is the prerequisite for efficient interferon-stimulated gene (ISG) induction upon HIV-1 infection, highlighting the importance of PQBP1 as a co-sensor of the cGAS–STING pathway and its emerging role in innate immunity [35]. We will describe the role of PQBP1 in innate immunity in more detail in a separate paragraph.

Dynamin-2 is a GTPase whose function is inhibited when PQBP1 binds to it. The PQBP1–dynamin-2 interaction is essential for the ciliary formation in neurons, emphasizing the role of PQBP1 in neurodevelopment [36].

Furthermore, the WWD binds to a non-structural protein of the avian reovirus (ARV) known as p17. ARV infection can cause severe symptoms in poultry, and p17 plays a role in the viral replication of ARV [38,39,40,41]. There is a reciprocal effect of PQBP1 expression and ARV replication which makes PQBP1 an interesting host factor for ARV infection [38,39]. The role of PQBP1 in the context of ARV infection will be discussed later.

The PRD (residues 104–163) is a largely unstructured protein motif that contains hepta- and di-amino acid repeats and interacts directly with polyglutamine (polyQ) tracts. PQBP1 was first identified as a protein that has the ability to bind to the polyQ tract of transcription factor Brn-2 [6]. Brn-2 is a transcription factor mainly expressed in the central nervous system (CNS) that is crucial for the development of cognitive functions and hippocampal neurogenesis [42]. PQBP1 inhibits Brn-2’s ability to activate gene transcription [6].

Interestingly, Waragai et al. also found that PQBP1 binds to polyQ tracts of proteins associated with neurodegenerative polyQ diseases, including mutated huntingtin (Htt), related to Huntington’s disease, and mutated androgen receptor, which is linked to Kennedy’s disease [6]. These diseases are associated with a pathogenic expansion of genetic repeats encoding stretches of glutamines in the disease-associated proteins [43]. Similarly, PQBP1 was found to bind to mutated ataxin-1, known to be in association with spinocerebellar ataxia type 1 (SCA1). This interaction reduced the levels of phosphorylated RNA polymerase II, repressed transcription and induced cell death [12].

Furthermore, such large unstructured polar-amino-acid-rich domains with low complexity have been found in many proteins involved in liquid–liquid phase separation (LLPS) [44]. In recent years, many research projects have aimed to show that LLPS can orchestrate intracellular compartmentalization. Particularly, Yang et al. showed clearly the involvement of PQBP1 in phase separation and how this peculiar behavior can be affected by mutations in the PRD and CTD domains of PQBP1 [45].

Like the PRD, the NLS (residues 170–187) is also composed of an unstructured proline–tyrosine-rich domain that allows karyopherin β2 (Kapβ2) to bind [46,47], which mediates the nuclear import of bound cargo [46]. Liu et al. described that PQBP1 binds both Kapβ2 and other proteins simultaneously, and thereby facilitates a piggyback mechanism for those proteins to enter the nucleus [47]. In fact, that mechanism was suggested to be one of the nuclear import routes of WBP11 [31,47]. In line with this, PQBP1 can be found both in the nucleus and in the cytoplasm [6,7,48]. Liu et al. identified U5-15kD as another protein that uses PQBP1 for nuclear import [47].

U5-15kD is a spliceosomal protein that binds to a 23-residue segment in the CTD of PQBP1 (residues 190–265) [49]. Mizuguchi et al. showed that PQBP1 forms a complex with U5-15kD and U5-52kD that is in turn incorporated into the spliceosome [33]. They also demonstrated, providing a crystal structure of the complex (PDB 4BWD, 4BWS, 4CDO), that the YxxPxxVL motif in the CTD of PQBP1 is crucial for the formation of the complex with U5-15kD and U5-52kD and the stabilization of the spliceosome. Interestingly, most disease-associated mutations within the Renpenning syndrome spectrum lead to frameshift mutations, resulting in an altered amino acid sequence and truncation of the CTD. This goes hand in hand with the loss of the YxxPxxVL motif [33]. Furthermore, Courraud et al. recently analyzed the consequences of PQBP1 deficiency in human neural stem cells, reporting a high score enrichment and deregulation of genes well known to be involved in neurodevelopmental and mitochondrial disorders (e.g., APC2, MT-CO2) and RNA metabolism [50]. In particular, they identified UPF3B_S, a short isoform of UPF3B with unknown function, as a good biomarker with higher expression in the blood of individuals harboring PQBP1 CTD mutations, particularly for R243W, P244L and F240fs, suggesting that the CTD might have a strong effect in the regulation of this gene [50]. Notably, the variant R243W has been found recently in a Turkish patient showing hypogammaglobulinemia, a phenotype never seen before in the Renpenning syndrome spectrum [51].

Post-translation modifications in the CTD (e.g., phosphorylation on serine-247) seem to be important for the regulation of the immunostimulatory activity. He et al. showed in 2022 that LATS2, a core component of the Hippo signaling pathway, which can be activated by a number of signals such as hormones or stress, phosphorylates PQBP1. The phosphorylation enhances the binding of PQBP1 and cGAS and the oligomerization of cGAS, leading to an enhanced innate response upon HIV-1 infection [52]. Based on these findings, the authors hypothesized that LATS2 phosphorylates YAP/TAZ to inhibit cell proliferation and phosphorylates PQBP1 to promote the activity of cGAS [52]. This establishes a new role of LATS2, strengthening the connection between the Hippo pathway and innate immune signaling. The components of the Hippo signaling pathway have been implicated in innate immunity before [53]. For instance, YAP and TAZ were found to suppress TBK1, a key player in the cGAS–STING signaling pathway. Upon LATS1/2 activation, YAP and TAZ no longer suppress TBK1 [54].

In another study, a truncation analysis of PQBP1 with subsequent co-immunoprecipitation analysis verified that amino acids 104 to 211 (including PRD, NLS and the start of the CTD) are needed for cGAS binding [7]. The previously identified cGAS binding site in the WW domain, which is mutated in Golabi–Ito–Hall syndrome (Y65C), is by itself not sufficient for the interaction with cGAS [7,35].

Yoh et al. also demonstrated specific enrichment of lentiviral-derived DNA after pulldown of PQBP1 from infected cells in comparison to control pulldowns, even in the absence of cGAS upon silencing. This enrichment was reduced when PQBP1 was truncated in the CTD, suggesting binding of lentiviral DNA products to CTD [35]. The authors also discovered that PQBP1 NTD and CTD act cooperatively: cGAS and capsid bind distinct surfaces on PQBP1, respectively, and secondly, there is a temporal component, first binding HIV-1 capsid, the latter triggering cGAS activation, suggesting potential structural changes [7].

These results particularly emphasize the key role of PQBP1 in the activation and regulation of innate immunity.

### 2.2. Additional Protein Interactions

Many other interactions with PQBP1 have been described in literature that have not yet been assigned to a specific domain. In 2004, Makarova et al. showed that PQBP1 and WBP11 associate with the CDC5/Prp19 complex, which is recruited to the spliceosome to form the pre-catalytic B complex. After the catalytic activation, PQBP1 and WBP11 no longer associate with the spliceosome. Moreover, they hypothesized a role for PQBP1 and WBP11 as a docking site for the CDC5/Prp19 complex at the spliceosome [55]. The association of PQBP1 with U5-15kD, which is also part of the spliceosome, supports this theory [49].

Kunde et al. described an association of PQBP1 with RNA. Their research showed that PQBP1 forms a complex with the cytoplasmic proteins K-homology-type splicing regulatory protein (KSRP), polypyrimidine tract-associated splicing factor (PSF), cell cycle-associated protein 1 (Caprin-1) and DEAD box polypeptide 1 (DDX1). The complex of these cytoplasmic proteins and PQBP1 associates with both RNA granules and the dynactin complex that is involved in the transport of organelles across microtubules. The authors hypothesize that PQBP1 facilitates the association of RNA granules and motor proteins, thereby allowing the movement of RNA granules throughout the cell [48].

Interactions have been reported with regard to intellectual disability (ID) and impaired neurodevelopment. In 2013, Goh et al. reported that both the metabotropic glutamate receptor (mGluR) and N-methyl D-aspartate (NMDA) receptors are necessary for learning behavior. Antagonism of either receptor led to impaired hippocampal long-term depression (LTD). LTD is crucial for synaptic plasticity that allows learning behavior and memory [56]. Shen et al. investigated an important role for PQBP1 inmGluR signaling. Upon stimulation, mGluR induces activation of eEF2K, which in turn phosphorylates eEF2. Phosphorylated eEF2 stimulates the translation of Arc [4]. Translation of Arc is implicated in LTD and synapse elimination, which are both processes that are crucial to learning behavior and memory [57]. This signaling pathway is called the mGluR-LTD and has been associated with multiple neurological disorders including Alzheimer’s disease (AD). Shen et al. showed that cytosolic PQBP1 binds to unphosphorylated eEF2, thereby preventing its phosphorylation and inactivation [4]. Counterintuitively, the authors also showed that PQBP1-mediated suppression of eEF2 phosphorylation is necessary for the induction of the mGluR-LTD by de novo translation of Arc. In conclusion, PQBP1 expression is necessary for proper learning behavior and memory. These findings are in line with results of in vivo experiments on the function of PQBP1 [58,59,60]. These observations indicate glutamate receptor signaling as an explanation for intellectual disability (ID) and impaired neurodevelopment mediated by mutated PQBP1, and might explain the phenotype which is often observed in Renpenning syndrome spectrum patients. Additionally, a protein–protein network analysis revealed that anaphase-promoting complex subunit 4 (Apc4) was linked to Pqbp1 via U5-15kD. In vitro validation experiments revealed that PQBP1 is essential for *Apc4* transcription and pre-mRNA splicing. In conclusion, PQBP1 depletion in mice delayed cell cycle time through Apc4 and likely other molecules that were aberrantly transcribed or spliced, which causes microcephaly [61]. With this mouse model, the researchers showed that PQBP1 may be responsible for the symptoms seen in Renpenning syndrome spectrum patients and the single causative gene of these disorders.

## 3. PQBP1 Role in Innate Immunity

The multifunctional nature of PQBP1 was emphasized in 2015, when Yoh et al. first described a role for PQBP1 in innate immune signaling [35]. The innate immune system is equipped with pattern recognition receptors (PRRs) that serve to recognize pathogen-associated molecular patterns (PAMPs) [62]. cGAS has been identified as the main PRR sensing cytosolic DNA. Upon recognition, cGAS synthesizes 2′,3′-cGAMP as a second messenger which activates the stimulator of IFN genes protein (STING) [63,64]. This activates the cGAS–STING pathway that finally results in type I IFN, IFN-stimulated gene (ISG) and cytokine expression, and thereby activation of defense mechanisms [65].

In 2015, Yoh et al. [35] reported that activation of the cGAS–STING pathway upon HIV-1 infection was mediated by PQBP1. Renpenning syndrome patients’ monocyte-derived dendritic cells harboring disease-associated mutations displayed a severe impairment in the innate immune response to HIV-1 infection. The dependence on PQBP1 for the innate response was common to other lentiviruses, including feline immunodeficiency virus (FIV) and equine infectious anemia virus (EIAV). In absence of PQBP1, the innate reaction was distinctly reduced upon lentiviral infection, whereas the immune response towards a dsDNA-virus infection by MHV-68 or transfected DNA (B-DNA and HT-DNA) was not affected by depletion of PQBP1, suggesting a specific response to retroviral PAMPs. Upon further investigation, Yoh et al. found that PQBP1 directly associates with cGAS through its WW domain and that the CTD is required for binding to HIV-1 reverse-transcribed DNA, which regulates the activation of cGAS during lentiviral infection [35]. These findings indicate that PQBP1 is required for the efficient detection of low-abundant and transiently produced PAMP generated during early-stage lentiviral infection. The same group recently elucidated the intricate details of the PQBP1-mediated cGAS activation. Upon HIV-1 infection, PQBP1 associates with a positively charged arginine-18 ring within the central channel of CA hexamers of the HIV-1 capsid in the cytoplasm [7]. Interestingly, the capsid structure that PQBP1 recognizes is highly conserved in the lentivirus genus. Recently, the interaction between PQBP1 and the HIV-1 capsid was further elucidated. Piacentini et al. confirmed that PQBP1 binds to the arginine-18 ring in CA hexamers and that this interaction appears by charge complementing contacts with acidic amino acid residues in the PQBP1 NTD [66]. In addition, they suggest that this is the primary interaction of PQBP1 and HIV-1 capsid, which is then followed by supplementary elements providing a stable binding [66].

Yoh et al. demonstrated the association of PQBP1 with the capsid to be the initial step of a two-step innate immune recognition of reverse-transcribed HIV-1 DNA encoding a temporal component. First, intact capsids of incoming HIV-1 are decorated with PQBP1. This is the prerequisite for the recruitment of cGAS by PQBP1 upon capsid disassembly and nascent HIV-1 DNA generation, leading to enzymatic activation of cGAS measured by cGAMP production. This mechanism ensures that transient and low-abundant reverse-transcribed HIV-1 DNA early upon infection in the cytoplasm will be localized in proximity to the cGAS receptor. Truncation of the C-terminal domain of PQBP1 led to loss of cGAS binding, suggesting distinct binding surfaces on PQBP1 for capsid and cGAS, respectively. Furthermore, the sequential recruitment and different binding surface of cGAS recruitment to the PQBP1–capsid platform suggests that conformational changes or other molecular signals, such as post-translational modifications, might be essential. Interestingly, HIV-1 infection effectively enhanced PQBP1–cGAS complex formation, in stark contrast to HT DNA-treated cells, further supporting the specificity for lentiviral recognition in the cytoplasm to trigger a specific antiviral response, rather than an unwanted response to host-derived DNA possibly leaking from mitochondria or the nucleus (see Figure 2B) [7].

This theory was supported by findings of Shannon et al. in which PQBP1-deficient cells attenuated the IFN response against EIAV lentiviral vector [67]. However, after transfecting DNA encoding an HIV sequence motif into PQBP1-deficient THP-1 cells, the authors found that type I IFN production was unexpectedly increased compared to cells expressing PQBP1, indicating an inhibitory role of PQBP1 on the cGAS-induced interferon response. Similarly, bacterial genomic DNA, poly(dA:dT) and damaged host DNA induced a stronger IFN response in PQBP1-deficient cells than in control cells [67]. The negative regulatory role of PQBP1 in innate sensing might be blocked by capsid binding through lentiviral vectors or viruses, which then enables PQBP1 to recruit cGAS to the nascent DNA supported by the two-step authentication mechanism demonstrated by Yoh et al. [7].

The observation that PQBP1 can inhibit IFN production in response to cytosolic DNA poses an interesting question on its role within the cGAS–STING pathway. As mentioned before, DNA from different origins can activate the cGAS–STING pathway. The finding that PQBP1 can inhibit the pathway in response to cytosolic DNA in the absence of lentiviral capsid hints at an additional regulatory mechanism between PQBP1 and cGAS that is not yet uncovered [67]. Future research could reveal how PQBP1 can both activate and inhibit the cGAS–STING pathway in a capsid-dependent manner.

PQBP1 also was identified to play a role in innate immune response induced by ARV. ARV is a non-enveloped virus belonging to subgroup II of the genus Orthoreovirus of the Reoviridae family. It carries a double-stranded RNA genome, made up of 10 segments encoding 10 structural and four non-structural proteins [68]. ARV infection can cause severe diseases in poultry, making it a threat for the poultry industry [69]. Through yeast two-hybrid screening, Zhang et al. identified PQBP1 to be a host factor interacting through its WWD with p17, one of the nonstructural proteins, a multifunctional protein involved in many cellular pathways [39,70]. PQBP1 mRNA expression was significantly decreased upon infection with ARV. Moreover, PQBP1 overexpression has an inhibitory effect on ARV replication, whereas PQBP1 knockdown increased ARV replication (see Figure 2A), suggesting PQBP1 as a restricting factor for ARV infection [39,70].

Investigation of the role of PQBP1 in innate immune response to ARV infection suggests that an inflammatory response towards ARV infection, such as IFN-β, IL-18 and Caspase-1 expression, is dependent on the presence of PQBP1. Furthermore, PQBP1 might be involved in NF-κB pathway induction upon viral infection, as Zhang et al. observed that p65 phosphorylation is induced by p17 or viral infection. They demonstrated that overexpressed PQBP1 increases even the level of this active form, whereas PQBP1 knockdown suppresses this promoting effect [70]. These results suggest that PQBP1 is involved in the innate immune response to different viral infections.

Morchikh et al. reported a complex around the protein hexamethylene bis-acetamide-inducible protein 1 (HEXIM1), a transcriptional inhibitor, and the interaction of the complex with cGAS and PQBP1. HEXIM1 can bind to the long non-coding RNA NEAT1 to form a platform for nuclear paraspeckle proteins SFPQ, PSPC1 and NONO as well as the DNA–PK complex [71]. The DNA–PK complex consists of the subunits DNAPKc, Ku70 and Ku80 and is itself known to activate the STING pathway in response to DNA [72]. The complex, called HDP–RNP, regulates the nuclear innate response upstream of IRF3 and TBK1 phosphorylation, but downstream of cGAMP synthesis, suggesting that this complex might not be directly involved in sensing of PAMPs. Stimulatory double-stranded (ds) DNA regulates the remodeling of interaction among HEXIM1, cGAS, PQBP1, STING and IRF3. Intriguingly, the complex is involved in the cGAS-mediated innate response against nuclear replicating Kaposi’s sarcoma-associated herpesvirus (KSHV). KHSV is well known to counteract the cGAS–STING pathway [73], for instance through the viral ORF52 protein. It prevents IRF3 phosphorylation and impairs the binding of cGAS and PQBP1 to HEXIM1 (see Figure 2D). This is particularly interesting in light of the fact that the authors identify HDP–RNP as an important regulator of nuclear ISD (interferon stimulatory DNA) and KSHV-mediated activation of the cGAS–STING pathway. Therefore, they may have elucidated a role for nuclear-localized PQBP1 as a supporting co-factor for the innate immune response. It is known that DNA-induced LLPS of cGAS is critical for cGAS enzymatic activity and downstream signal transduction [74]. Xu et al. found that ORF52 can target and restrict such cGAS–DNA condensates in early stages of KSHV infection, suggesting a new immune evasion strategy. In contrast to Wu et al., Xu et al. did not find a direct interaction between ORF52 and cGAS via isothermal titration calorimetry (ITC) or pull-down experiments [75]. However, they observed that ORF52 and DNA can form their own liquid-like droplets, which presumably leads to the collapse of previously formed cGAS–DNA phase separation. In addition, ATP and GTP, which normally also concentrate in cGAS–DNA droplets, were also released after addition of ORF52. This suggests that ORF52 has an effect of the accumulation of the enzyme itself but also on its substrates [75]. Interestingly, ORF52 displays an intrinsically disordered C-terminus which seems involved in phase separation formation.

Another tegument protein, VP22, encoded by herpes simplex virus type 1 (HSV-1) of the Alphaherpesvirinae subfamily [76] can form DNA-induced phase separation and restrict cGAS–DNA phase separation, similar to ORF52 from KHSV. Also for this protein, its N-terminal disordered region seems to play an important role in that context. Altogether, these data highlight the important roles of IDP regions in DNA-induced phase separation as well as restriction of cGAS–DNA phase separation.

Considering that PQBP1 is mainly composed of IDP regions and specifically required for inducing an innate immune response against lentiviral infection but can also inhibit IFN production in response to cytosolic DNA in the absence of a lentiviral capsid, one could speculate whether PQBP1 plays a role within LLPS, similarly to ORF52 [7,35,67]. Future research is needed to understand whether PQBP1 may be able to displace cGAS from LLPS as a negative regulator or whether PQBP1 may function in a supporting role during LLPS-mediated cGAS sensing. It is known that PQBP1 itself frequently occurs in such droplets [45].

Interestingly, HSV-1 is a major cause of CNS infections [77]. Type 1 IFNs are important for the control of HSV-1 in the CNS, and microglia are the major source of HSV-induced type 1 IFN expression in the CNS [78,79]. Here, IFNs are induced in a cGAS–STING-dependent manner [79]. Mice defective in cGAS or STING are rendered very susceptible to herpes simplex encephalitis, a form of acute viral encephalitis infection which causes irreversible damage to the CNS and fatality. HSV-1 infection could persistently activate PRRs, possibly including PQBP1, leading to chronic neuroinflammatory response in CNS and gradual neuron loss, which is a hallmark of neurodegenerative disorders [80].

Feng et al. recently reported that HSV-1 can worsen the pathogenesis of AD [81]. It is known that persistent inflammation in the CNS can be pathologic by causing exuberant cellular stress and cytotoxicity, especially in the brains of AD patients [81,82,83,84,85,86]. Not only herpesviruses, but many other viruses can establish infections in the CNS, such as flaviviruses or picornaviruses, that are known to trigger neuroinflammation [87]. HIV-1 also is known to enter the CNS via a “Trojan horse” mechanism via infected cells passing the blood brain barrier, infecting microglia, resident macrophages and possibly astrocytes. Inflammatory factors released by exposed or infected cells are implicated in damage of neurons, which leads to CNS pathology, a neuronal dysfunction known as HAND (HIV-1 associated neurocognitive disorders) [88]. With respect to HSV-1, neurons limit viral replication; however, only microglial cells (and to a lesser extent astrocytes) respond to HSV-1 infection in a cGAS–STING-dependent manner [79]. The involvement of PQBP1 in any of the aforementioned cell types and virus infections is still a matter for investigation. However, the role of PQBP1 in neurodegenerative disorders will be discussed in more detail in the next chapter.
Figure 2Role of PQBP1 in the innate immune response to different viral infections and disease-associated proteins. (**A**) PQBP1 interacts with the ARV nonstructural protein p17 [38]. There is a reciprocal relationship between PQBP1 expression and ARV replication. PQBP1 regulates expression of different inflammatory factors, which expression is upregulated during ARV infection. This might be related to the NF-κB pathway since p65 phosphorylation is induced by p17 and PQBP1 overexpression increases the level of this active form, whereas PQBP1 knockdown has the opposite effect during ARV infection [39]. (**B**) Upon cellular entry of HIV 1, (1.) the viral capsid is bound by PQBP1, which is the first step to initiate cellular sensing of HIV-1. (2.) Once capsid disassembly and HIV-1 DNA synthesis have been initiated, cGAS is recruited to the capsid in a PQBP1-dependent manner. This places cGAS at the site of PAMP formation, enabling sensing of temporally and spatially constrained HIV-1 DNA. This leads to cGAMP production and activation of downstream STING–IRF3 signaling [7]. (**C**) In microglia, PQBP1 recognizes tau protein, associated with neurodegenerative Alzheimer’s and Parkinson’s diseases. Binding of PQBP1 to tau induces the activation of the cGAS–STING pathway [14]. (**D**) PQBP1, as well as cGAS and IRF3, interact with the HDP–RNP complex. HDP–RNP regulates the induction of the cGAS–STING pathway by stimulatory dsDNA [71]. During infection of the herpesvirus KSHV, harboring a large dsDNA genome, KSHV protein ORF52 reduces the interaction of cGAS and PQBP1 to the HDP–RNP as well as IRF3 phosphorylation. The exact role of PQBP1 in this context is still under investigation.
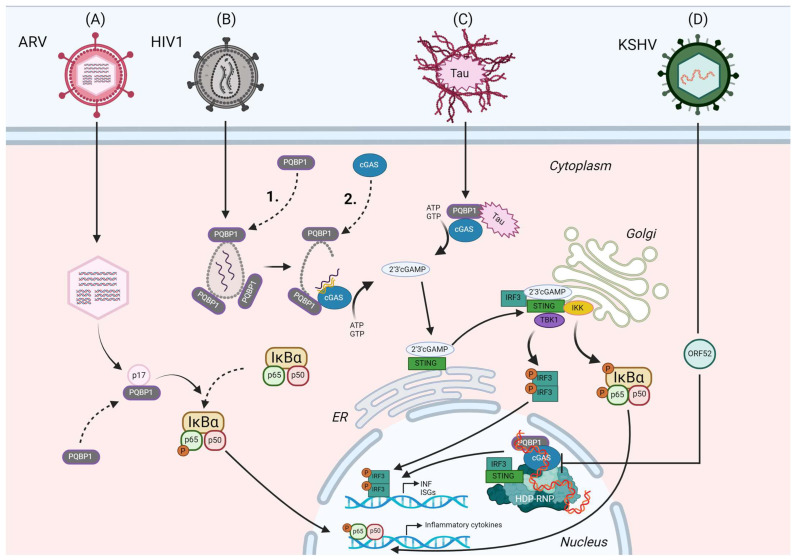



## 4. PQBP1 in Neurodegenerative Disorders

Progressive loss of neurons due to protein aggregate deposits and altered physicochemical properties in the brain are the common hallmarks of neurodegenerative disorders. Intracellular damage and activation of neuroinflammation are the driving forces for progressive neurodegeneration [89].

It is already widely reported in the literature that conformational changes and misfolding of several IDPs can produce toxic accumulation and spreading in a “prion-like” behavior of oligomeric species, leading to neurodegeneration. However, the next goal for research in the field would be to highlight the presence of common features in these disorders. Due to the direct correlation with neurodevelopmental diseases and its role in innate immunity, PQBP1 is one of these candidates [90]. Jin et al. showed that in microglia, PQBP1 recognizes extrinsic tau protein, which has been associated with several neurodegenerative disorders including AD, frontotemporal dementia with parkinsonism-17 (FTDP-17), Pick disease (PiD), progressive supranuclear palsy (PSP) and corticobasal degeneration (CBD) [91]. Pathological tau proteins can be secreted and internalized by neighboring cells. Although the mechanism of extracellular tau internalization is not clear, several possible mechanism, such as phagocytosis, pinocytosis and mediated uptake are proposed [92]. The authors propose that tau interacts with PQBP1 since cytoplasmic tau foci colocalize with PQBP1, which drives innate immune responses. Activation of the cGAS–STING pathway, translocation of NF-κB factor and expression of pro-inflammatory genes (e.g., *TNF, ISG54*) are shown to be PQBP1-dependent after uptake of tau proteins in microglial cells, eliciting neuroinflammation (see Figure 2C) [14]. These findings suggest a mechanism for PQBP1–cGAS–STING-mediated neurodegeneration in tauopathies that may also translate to other neurodegenerative diseases in which PQBP1 interacts with the causative disease protein.

Furthermore, PQBP1 has been implicated in other neurological conditions collectively called the triplet repeat diseases. Triplet repeat diseases are neurodegenerative disorders that are associated with an expansion of a trinucleotide repeat sequence in causative genes, resulting in expanded polyglutamine tracts in the gene products [93]. PQBP1 is well known to interact with polyQ sequences through its PRD. Abnormally expanded forms of these polyQ tracts could even lead, through PQBP1, to a first cellular innate response against these pathogenic proteins. Indeed, PQBP1 can bind to a number of causative disease proteins, such as mutated huntingtin associated with Huntington’s disease as well as the mutated androgen receptor that is associated with Kennedy’s disease [6,13]. Another study reported that PQBP1 binds to ataxin-1, the disease protein of SCA1, and that the binding affinity is even higher for mutated ataxin-1 with an abnormally elongated polyQ tract. When PQBP1 binds to mutant ataxin-1, PQBP1 binding to RNA polymerase II (Pol II) is enhanced. This prevents the phosphorylation of PolII, and thereby represses transcription, since PolII stays in the unphosphorylated, inactive, form. Moreover, the interaction between mutated ataxin-1, PQBP1 and PolII is required for ataxin-1-mediated cell death [12].

As widely reported in the literature, implication of immunity in neurodegenerative disorders [94], aging and age-related disease [95] is directly connected either to mitochondrial damage or to cytosolic chromatin fragments [96] in senescent cells. Mitochondrial DNA (mtDNA) leakage has been found to be one of the major triggers for inflammatory response in aged microglia in a mouse model [97], explaining the upregulation of interferon genes in several tissues [98]. All these results clearly propose the DNA sensor cGAS as a critical player in chronic inflammation and tissue degeneration, in which case PQBP1 might act as a regulator of cGAS activity. As reported by Du and Chen [74], DNA–cGAS condensate formation seems to be crucial for the downstream signaling cascade. PQBP1 might also affect LLPS formation and could be an important player for these macromolecular condensates, boosting the innate response against retroviruses (HIV-1) or protein-based pathogens such as tau or mutated Htt.

## 5. Discussion

Its versatility is one of the most peculiar characteristics of PQBP1. The high number of proteins it can interact with, the direct interaction and involvement in pre-mRNA splicing as well as transcription make PQBP1 an important factor for cellular regulation and development. Since PQBP1 is an IDP, lacking secondary and tertiary structure despite the well-conserved WW domain [19], it is quite difficult to obtain molecular insights. Intriguingly, the presence of large low-complexity domains and high flexibility play a role in the formation of LLPS, which influences various cellular processes including innate immunity. Further investigation on PQBP1 as part of this macromolecular machinery is crucial to understand the role of PQBP1 in cellular processes.

Although the molecular mechanisms have not been completely elucidated, different mutations in distinct domains of PQBP1 are the hallmark of intellectual disability and development impairment resulting in Renpenning syndrome spectrum. Moreover, by assessing Renpenning-syndrome-spectrum-associated PQBP1 mutations, the importance of intact PQBP1 in the innate immune response to HIV-1 has been unraveled [35].

Indeed, PQBP1 can act as a central mediator in innate immunity, necessary for recognizing pathogens and recruiting cGAS for pathway activation. During lentiviral infection, the viral capsid is bound by PQBP1, which mediates a two-factor authentication strategy for innate sensing. cGAS is then recruited to the capsid and recognition of reverse-transcribed viral intermediates activate the cGAS–STING pathway [7]. This makes PQBP1 an important host factor in innate immunity, allowing for recognition of temporally and spatially limited lentiviral PAMPs in the cytosol. Moreover, recent studies further investigating the role of PQBP1 propose that the involvement of PQBP1 in innate immunity could be extended. Viral proteins can augment (ARV, p17) or inhibit (KSHV, ORF52) complexes containing PQBP1 [39,71]. The latter complex might be important for nuclear viral immunity. Furthermore, PQBP1 might act as a negative regulator of innate immunity [99]. Further research is needed to understand the separate functions of PQBP1 in the cytoplasm and nucleus.

PQBP1 is able to recognize disease-related proteins, such as tau protein, eliciting neuroinflammation and, subsequently, neuronal cell death [14]. It was previously described that PQBP1 is involved in the binding of polyQ-expanded tract proteins in polyQ diseases. However, it is unknown whether PQBP1-mediated chronic neuroinflammation contributes to later neurodegeneration. As discussed in this review, PQBP1-mediated neurodegeneration could be hypothesized in tauopathies such as AD, FTDP-17, PiD, PSP and CBD. Therefore, these results may be also translated to other neurodegenerative diseases in which PQBP1 interacts with the causative disease protein or in the context of infection by viruses known to infect the CNS or viruses that possibly pass the blood brain barrier.

Ultimately, the results discussed above suggest that a deeper characterization of PQBP1 interactome may provide valuable insights, unravelling new molecular mechanisms related to cellular regulation in innate immunity and neurodegeneration. With this review, we aim to provide a platform for further research into cellular functions of PQBP1.

## Figures and Tables

**Figure 1 viruses-16-01340-f001:**
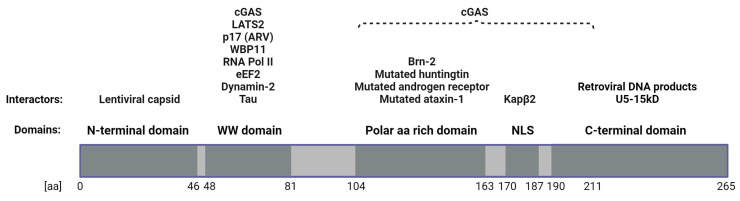
A schematic overview of the PQBP1 domains with their mapped interactors. PQBP1 comprises 265 amino acids, and five different domains have been identified. For each of these domains, specific interaction partners have been confirmed. Regarding the role of PQBP1 in innate immunity, of particular interest are its lentiviral capsid binding site in the NTD, the binding site for retroviral DNA products in the CTD as well as two cGAS binding sites in the WW domain and in a C-terminal region of the protein (residues 104–211). In terms of the proteins’ role in neurodegeneration, its binding sites for tau, mutated huntingtin, mutated androgen receptor as well as mutated ataxin-1 in the WW domain and the PRD are especially noteworthy.

## Data Availability

No new data were created.

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
