# Peer review of "Role of PQBP1 in Pathogen Recognition—Impact on Innate Immunity"

_viruses, 2024, doi:10.3390/v16081340_

Round 1
Reviewer 1 Report
Comments and Suggestions for Authors
This is a nice review of PQBP1 and summarizes essential information for readers who are interested in PQBP1. Meanwhile, I have several minor points that should be corrected before final decision of this submission.
1) It is necessary for authors to declare whether they used AI for generation of this manuscript, because this review is so well written.
2) Line 15-16. “remain unknown” sounds inappropriate because a large amount of data has been accumulated regarding “molecular mechanism of PQBP1”, as the authors wrote in this manuscript. Some other descriptions such as “remain to be further investigated” would be better.
3) Line 24-36. The authors described PQBP1 mutation at first, and then described PQBP1. This is contradictory to the history of PQBP1. PQBP1 was originally discovered as a binding protein to polyQ tract sequences, and then whose mutations were determined to be causative for intellectual disabilities. It is better to edit this paragraph. Also in the revision of this paragraph, the authors should refer the following paper as the original paper of PQBP1.
Imafuku I, Waragai M, Takeuchi S, Kanazawa I, Kawabata M, Mouradian MM and Okazawa H. Polar amino acid-rich sequences bind to polyglutamine tracts. Biochem Biophys Res Commun. 1998 Dec 9;253(1):16-20. doi: 10.1006/bbrc.1998.9725. PMID: 9875212.
4) Line 65. The period position is incorrect.
5) Line 87-89. Here the authors mention about splicing factors, and refer only U2 factors though they described U5 factors in the later sections. It is better to touch also about U5 factors very briefly here.
Reviewer 2 Report
Comments and Suggestions for Authors
Major deficiencies.
-Interactions of PQBP1 with viruses relevant to CNS virus infection is clearly underdeveloped for a meaningful review in a journal like “Viruses”.
-Overall, there is a superficial description of normal PQBP1 functions in various relevant CNS cells that play a role in neurodegenerative diseases. As such, there was a lot more information on the role of PQBP1 in basic developmental neuroscience than on the virology of PQBP1 in this review making the subject a bit out of scope for publishing in this journal.
-CNS cell specific function of PQBP1 is not meaningfully presented and discussed especially in the context of virus infections in the CNS.
-The authors discuss tau interactions of PQBP1 with cGAS/STING in the cytoplasm in the text but then emphasize extracellular tau in Figure 2. The reader is left with wondering about the relationship between extra-and intracellular tau and how this may affect eventual pathogenic interactions with PQBP1 in the cytoplasm. What is the role for altered cGAS/STING in neurodegenerative diseases? Meaningful hypotheses dealing with these interactions should be presented and discussed along with supportive citations.
-While implied, there is no thoughtful discussion of how viruses may promote neurodegenerative diseases via alterations in or interactions with PQBP1 either by affecting innate immunity or promoting other cellular changes in specific CNS neurons and/or glia.
Minor deficiencies.
-Sentence structure is repeatedly awkward.
-Many grammatical and spelling errors.
Comments on the Quality of English LanguageMany sentences are poorly written.
Many grammatical and spelling errors.
Round 2
Reviewer 2 Report
Comments and Suggestions for Authors
-The manuscript is much improved from the original especially with regard to elaborating the multiple interactions of PQBP1 with both viruses and innate immune signaling pathways.
Comments on the Quality of English Language
-Minor editing of english is still needed to maximize impact of the review.
Author Response
-The manuscript is much improved from the original especially with regard to elaborating the multiple interactions of PQBP1 with both viruses and innate immune signaling pathways.
Thank you for your suggestions which greatly improved the manuscript.
Comments on the Quality of English Language
-Minor editing of english is still needed to maximize impact of the review.
We asked a native English speaker to revise our manuscript and carefully edited the text.